# Development of a Tool for Determining the Equivalence of Nutritional Supplements to Diabetic Food Exchanges

**DOI:** 10.3390/nu14163267

**Published:** 2022-08-10

**Authors:** Paraskevi Detopoulou, Georgios I. Panoutsopoulos, Garifallia Kalonarchi, Olga Alexatou, Georgia Petropoulou, Vasilios Papamikos

**Affiliations:** 1Department of Clinical Nutrition, General Hospital Korgialenio Benakio, 11526 Athens, Greece; 2Department of Nutritional Sciences and Dietetics, University of the Peloponnese, 24100 Kalamata, Greece

**Keywords:** supplements, energy, protein, menu planning, food exchanges

## Abstract

Malnutrition is a prevalent issue in hospitals, nursing homes and the community setting. Nutritional products can be used by dietitians to supplement people’s diet by adding energy, macronutrients and other constituents. The aim of the present work was (i) to create a database of nutritional products with information on their energy and macronutrient content, (ii) to estimate the food exchanges of each product and assist in diet plan development for malnourished persons and (iii) to provide a tool for calculation of food exchanges of newly developed products not included in the database. We searched the web for nutritional supplements, and an electronic database with 461 products was generated with data regarding the contained energy and macronutrients of each entry. The following companies were included: Abbott Nutrition, Nestle Nutrition, Nutricia North America, Nutricia Global, Nutricia Europe & Middle East, Axcan Pharma Inc., Kate Farms, Global Health, High Protein, NutriMedical BV, Hormel Health Labs, Hormel Health Labs/Diamond Crystal Brands, Lyons Magnus, Mead Johnson, Medical Nutrition USA Inc., Medtrition, Nutritional Designs Inc., Nutrisens, Humana (Germany), and Vitaflo USA. The created database facilitates product comparisons and categorization into several groups according to energy and protein content. In addition, a tool was created to determine food exchanges for each supplement per serving and/or food exchanges for newly developed products by simply inserting their macronutrient content. The developed tool can facilitate dietitians in comparing products and incorporating them into diet plans, if needed. Such tools may thus serve clinical practice, may be used in dietary or other smart applications and can familiarize dietitians with the digital epoch.

## 1. Introduction

Malnutrition is an evident problem in hospitalized patients [1], nursing homes [2] and the community setting [3], affecting physical performance [4], mortality [5] and quality of life [6]. Possible dietary interventions include oral nutritional supplements (ONSs) along with dietary advice [7]. For example, ONSs and/or products with omega-3 fatty acids may be effective in reducing sarcopenia [8]. Indeed, in various diseases where protein intake is important, their use is recommended [9], while the European Society for Clinical Nutrition and Metabolism (ESPEN) recommends ONSs for older persons with malnutrition or at risk of malnutrition and chronic conditions if counseling and food fortification are insufficient [10].

ONSs are “dietary foods for special medical purposes” and are available as ready-made cream-like preparations, juice-type preparations, puddings and powders [11,12,13,14]. Liquid ONSs are also called sip feeds [14], and they may be nutritionally complete (covering 100% of the dietary recommendations) or nutritionally incomplete (adapted to contain specific nutrients such as carbohydrates, lipids or proteins in high concentrations) [14]. In every case, a “food first” approach should be followed through dietary counseling [10], and “tricks” to boost the energy and protein content of real foods and meals should be recommended [15,16,17]. If this is not possible, ONSs as part of dietary advice (under medical supervision) may have favorable effects on malnourished individuals, leading to weight gain [7] and improved quality of life [5].

As part of the nutrition care process, dietitians make personalized nutrition plans, considering dietary needs, preferences and health issues. Useful tools for menu planning are food exchange lists commonly used by dietitians [18]. These lists were initially developed by dietitians and diabetes health professionals in 1950 [19]. They represent groups of foods with a detailed description in terms of weight and type that approximately contribute the same grams of carbohydrates, fats, proteins and kilocalories per choice. Foods belonging to the same group are “interchangeable” since they can replace each other in dietary schemes. Several updates have been made to the exchange lists since their conception [18,20,21]. Moreover, additional versions of food exchanges lists have been developed for renal diseases [22] or to incorporate baby foods [23,24], ethnic foods [25,26,27], Mediterranean foods [28,29], vegetarian foods [30,31] or sports foods [31] in order to cover specific needs and facilitate meal planning.

To our knowledge there are no existing food exchanges lists for ONSs, and the existing algorithms may be time-consuming if performed on a per patient/client basis [18]. In addition, there is no large at-one-glance open-access ONS database available in the literature. The USDA database contains several products [32]; the US Dietary Supplement Ingredient Database includes multivitamin/mineral products, omega-3 supplements and botanicals [33]; and the Australian AUSNUT 2011–13 dietary supplement database includes ~2160 products, but none of them are ONSs (most are vitamins, fish oils, minerals, etc.) [34].

The aim of the present study was to create a tool for dietitians and health professionals that (i) includes a database of nutritional products with information on their energy and macronutrient content, (ii) serves in estimating the food exchanges of each product and can assist in developing a diet plan for malnourished persons and (iii) serves for calculation of food exchanges of newly developed products not included in the database so that every present or future need can be covered.

## 2. Methods

### 2.1. Oral Nutritional Supplement Database

The authors considered the oral nutritional supplements proposed by the American Academy of Nutrition and Dietetics and searched for products circulating in Europe. Inclusion criteria were: oral nutritional supplement containing macronutrients and availability of the macronutrient content from the company or the web. Exclusion criteria were: products designed for parenteral nutrition and products including only micronutrients. Then an electronic database was created with information on the energy and macronutrient content of 464 products. The following companies were included: Abbott Nutrition (Lake County, IL, USA), Nestle Nutrition (Vevey, Switzerland), Nutricia North America (Rockville, MD, USA), Nutricia Global (Utrecht, The Netherlands), Nutricia Europe & Middle East (Utrecht, The Netherlands), Axcan Pharma Inc. (Mont-Saint-Hilaire, QC, Canada), Kate Farms (Santa Barbara, CA, USA), Global Health (Rochester, NY, USA), High Protein (Aix-en-Provence, France), NutriMedical BV (Nootdorp, The Netherlands), Hormel Health Labs (Austin, MN, USA), Hormel Health Labs (Austin, MN, USA)/Diamond Crystal Brands (Atlanta, GA, USA), Lyons Magnus (Fresno, CA, USA), Mead Johnson (Chicago, IL, USA), Medical Nutrition USA Inc. (Englewood, NJ, USA), Medtrition (Lancaster, PA, USA), Nutritional Designs Inc. (Lynbrook, NY, USA), Nutrisens (Francheville, France), Humana (Bremen, Germany) and Vitaflo USA (Alexandria, VA, USA).The created list was not meant to be exhaustive since the number of newly developed products around the globe is increasing. All information was inserted in a Microsoft Excel file to increase visibility and data processing capacity for non-IT specialists (please see Appendix A, in an Excel format).

### 2.2. Categories of Oral Nutritional Supplements

A formula catalog with the total supplements was created and then different subcategories were formed according to proposed literature cutoffs adopted by ESPEN and other researchers, i.e., low energy < 1.0 kcal/mL, normal energy 1.0–1.2 kcal/mL, high energy > 1.2 kcal/mL, high protein > 20% energy from proteins [35,36]. Following these criteria, the oral nutritional supplements were described as follows: isocaloric supplements (with normal energy values) (*n* = 150), high-energy supplements (*n* = 304), high-energy–high-protein supplements (*n* = 302), high-energy–normal-protein supplements (*n* = 303), and high-protein supplements (*n* = 173). It is noted that some supplements may fall into more than one category.

### 2.3. Algorithm for Food Exchanges Calculation Tool

The published methodology by Wheeler et al. [18] was considered in order to determine food exchanges for each product. For a starch exchange, the product should contain 11–20 g of carbohydrates per serving. If it contained 1–5 g of carbohydrates, no carbohydrate exchange was calculated, while if it contained 6–10 g of carbohydrates, it was considered as half an exchange. If the content of dietary fiber was more than 5 g, only half of the fiber grams were counted in the total carbohydrate content. As far as protein choices are concerned, if the product had 4–10 g of protein per serving, it was considered as one exchange, while if it had less than 4 g, no protein exchange was considered. If the product had 4–7 g of fat per serving, a fat exchange was considered. If it contained less than 2 g of fat, no fat exchange was counted, whereas if it had 2–4 g of fat, half an exchange (fat choice) was considered.

## 3. Statistical Analysis

For normality testing, the Kolmogorov–Smirnoff test was used. Non-normally distributed continuous variables are shown as median and interquartile range. It is noted that supplements with high volumes (>500 mL) were excluded from the median (interquartile range) calculations of each supplement category. The Mann–Whitney non-parametric test was used to compare variables between isocaloric supplements and high-energy, high-energy–high-protein, high-energy–low-protein and high-energy–normal-protein supplements. No other comparisons were possible due to the overlapping of supplement categories. Since data were skewed, the Spearman correlation coefficient was used to test the associations of carbohydrate, fat and protein content with the calculated food exchanges. All reported P values are compared to a significance level of 0.05. IBM SPSS Statistics for Windows version 22.0 (Armonk, NY, USA: IBM Corp) software was used for all the statistical analysis. It is noted that for statistical comparisons, enteral nutrition products containing more than 500 mL were excluded from the analysis to obtain more homogeneous results (most products included were 130–300 mL).

## 4. Results

A total of 461 ONSs were analyzed, and the starch, fat and protein choices are provided per serving. The full database of supplements is included in Appendix A in an Excel format. In Table 1, the median (interquartile range) of macronutrients and corresponding food exchanges are presented for different categories (isocaloric, high-protein, high-energy, high-energy–high-protein, high-energy–normal-protein). Briefly, when considering the total database, a food supplement corresponds to 1.5 starch exchange, 1 protein exchange and 1 fat exchange (median values). For isocaloric supplements, the corresponding values of starch, protein and fat exchange are 1, 1 and 1 (median values), while in high-energy products they are 1.5, 1 and 1(median values). Single portions of high-energy and high-protein products correspond to 1 starch, 2 protein and 0 fat exchanges (median values); one portion of high-energy–normal-protein products corresponds to 2 starch, 1 protein and 2 fat exchanges (median values); and one portion of high-protein products corresponds to 1 starch exchange, 2 protein exchanges and 1 fat exchange (median values).

Figure 1 depicts the graphical association of supplements’ macronutrient content per portion (in grams) and food exchanges. As is clearly seen, a linear positive relation is observed between food exchanges and grams of macronutrients in the investigated products. Table 2 shows the Spearman correlation coefficients between macronutrient content per portion (in grams) and food exchanges. As is shown, the highest correlation coefficients were present for relevant macronutrient category and food exchange group (i.e., carbohydrate grams with starch exchanges, protein grams with lean meat exchanges and fat grams with fat exchanges). It is noted that the magnitude of correlations of fiber content with macronutrient content and food exchanges is lower (correlation coefficients 0.168–0.238). Table 3 represents an example of a supplement incorporation in a 2100 kcal meal plan for 70 kg, which has been designed by the food exchanges method. In this example, energy was calculated as 30 kcal/kg and protein was calculated as >1 g/protein/kg according to ESPEN recommendations for older adults [10]. As can be seen, by incorporating two cans of an ONS in a diet plan, the patient can obtain five choices of carbohydrates, two choices of lean meat and four choices of fat in a liquid, easy-to-consume form. The calorie and protein target can be thus reached more easily by supplementing the diet with ONSs.

A tool for calculation of food exchanges for other supplements or incoming products is provided in Appendix A in an Excel format. It is noted that the Excel file is locked (protected) and the only cells that can be filled in are I2, K2, M2 and O2 (with numbers). This has been considered necessary in order not to “intervene” in the embedded equations of the Excel file and provide correct calculations for the food exchanges.

## 5. Discussion

We present a nutritional supplement database with information on energy and macronutrient content as well as the corresponding food exchanges of each entry to assist in developing dietary schemes for malnourished people. Moreover, we provide an evidence-based tool to use for every other ONS not included in the database so that every present or future need can be covered.

It is noted that limited online calculators exist concerning formula calculations, such as ClinCalc.com, which, however, includes only ~50 products [37], and Nutritracker, recently introduced by NUMIL Hellas S.A. (Athens, Greece) [38]. Similarly, the USDA food database includes several ONSs in various categories such as “nutritional beverages”, “protein and nutritional powders”, “powdered drinks” and “milk additives”, which may render the identification and comparisons of products difficult [32].

In this study, we matched ONSs to starch, lean protein, and fat exchange values, based on their macronutrient content. We decided that the food exchanges should be as simple as possible. For example, we did not use milk exchanges which contain carbohydrates, protein and fat [18]. Although ONSs usually contain calcium, their macronutrient profile differs from that of milk, since in most cases they do not contain lactose and their main carbohydrate is maltodextrin [14]. The micronutrient content of the formulated diet with the use of ONSs is possibly different from the basic foods included in diabetic food exchange lists since ONSs are fortified products that in certain cases can fully cover the vitamin and micronutrient needs of a person. Of course, other combinations, i.e., the use of other food combinations, could be possible.

According to our results, an isocaloric supplement can provide 1 starch, 1 protein and 1 fat exchange, while high-protein supplements irrespective of their energy content provided 2 protein exchanges (higher median values than standard choices, as expected). High-energy choices had higher carbohydrate content corresponding to 1.5–2 starch exchanges, which means that the extra energy comes mostly from carbohydrates. This observation deserves attention for patients with diabetes. In other words, the prescription of a high-energy supplement implies a higher provision of a carbohydrate load which could affect postprandial glucose values, unless a specific carbohydrate type and/or fiber are present [39].

Within each food exchange group, one food exchange is considered approximately equal to the others as far as energy and macronutrients are considered and could be used “interchangeably” with other products. In this context, to enhance flexibility, several alternatives for the same dietary plan could be proposed by health professionals by using the provided data. The issue of having alternative choices in enteral nutrition products was recently underlined during the COVID epidemic, in which shortages in some products were observed [40]. Moreover, taste changes may be observed during treatment (such as radiotherapy) [41], and health professionals should be ready to propose alternative choices to their patients. Our provided database could help resolve such issues at a glance by providing choices with similar nutrient content and facilitating clinical practice.

Although macronutrient contents and corresponding food exchanges were highly correlated, matching of whole supplement categories to food exchanges should be interpreted with caution, since intragroup differences were large, supplement volumes differed and data were skewed. A proper approach when it comes to a patient-based basis would be to consider the food exchanges of the supplement that is going to be used by consulting the provided database or other available data. Moreover, detailed data on micronutrients were out of the scope of this work, and some products may contain immunoregulating substances (such as nucleotides, glutamine, arginine, omega-3 fatty acids and pre- and probiotics) [42] which give them an additive nutritional value compared to others, even if they “belong” to the same group. Other specific characteristics of several products (which are not included in our database) such as medium-chain fatty acids and carbohydrate profile should also be considered to address malabsorption issues and diabetes management, respectively [39], while the formula protein composition may affect gastric content volume [43].

The strengths of our study are that a relatively large supplement database was formed and an open tool for food exchanges calculation was provided for supplements included or not included in the database. An illustrative example is also provided to facilitate the incorporation of ONSs into meal plans.

The main limitation of this study is the restricted number of supplements included and the need for periodic updating due to new product development, product changes and product differentiations around the globe. Changes in products’ macronutrient content may take place, and most annotated data can be found on the manufacturer websites. However, the tool provided can be used for new data insertion and calculation of diabetic exchanges for practically every existing and upcoming product. The present list should be used with the aid of a dietitian or health professional, i.e., in conjunction with clinical judgment. In addition, after the prescription of an ONS, an understanding of the diet plan and adherence to it should be ensured. A “food first” approach should be initially used. Dietary counseling to encourage the use of energy- and protein-rich foods should be recommended as the initial intervention before prescribing ONSs, and thus ONSs are mostly intended to complement meal plans.

## 6. Conclusions

In conclusion, the presented data “made from dietitians for dietitians” offer a chance for dietitians and health professionals to efficiently incorporate ONSs in diet plans by calculating the starch, protein and fat choices of products. Our provided database can help in finding alternative ONS choices at a glance, facilitating clinical practice, and the proposed algorithm can help professionals calculate the corresponding food exchanges for ONSs not included in the database. The inclusion of ONSs may help in achieving the energy and nutrient goals of patients. The present work serves clinical practice; it provides a tool that can be used in dietary or other smart applications and can familiarize dietitians with the digital epoch.

## Figures and Tables

**Figure 1 nutrients-14-03267-f001:**
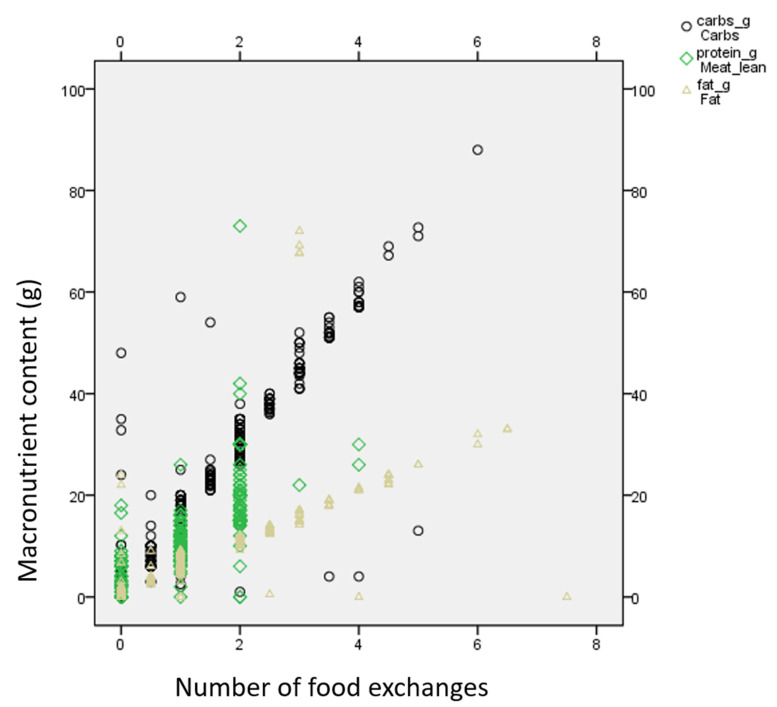
Scatter dot graph between grams of macronutrients and calculated food exchanges.

**Table 1 nutrients-14-03267-t001:** Macronutrients and diabetic exchanges per category of supplement.

	Total (*n* = 415)	Isocaloric Supplements (*n* = 123)	High-Energy Supplements (*n* = 292)	High-Energy–High-Protein Supplements (*n* = 114)	High-Energy–Normal-Protein Supplements (*n* = 178)	High-Protein Supplements (*n* = 161)
**Carbohydrate (g)**	21 (7.6–35.7)	19 ^b,c,d^ (8.5–32)	23 (7.4–37.1)	15 ^b^ (8–26)	29.5 ^c^ (7–44)	16 ^d^ (6–24.5)
**Starch exchanges**	1.5 (0.5–2)	1 ^b,c,d^ (0.5–2)	1.5 (0.5–2.5)	1 ^b^ (0.5–2)	2 ^c^ (0.5–3)	1 ^d^ (0.5–1.5)
**Protein (g)**	9 (4–15)	8 ^a,b,c,d^ (0–14)	10 ^a^ (6–16)	15 ^b^ (11–22)	7 ^c^ (0–11.9)	15 ^d^ (11–20)
**Protein exchanges**	1 (0–2)	1 ^a,b,c,d^ (0–1)	1 ^a^ (0–2)	2 ^b^ (1–2)	1 ^c^ (0–1)	2 ^d^ (1–2)
**Fat (g)**	5 (0–11)	2 ^a,b,c,d^ (0–6)	6.3 ^a^ (0–12)	1.55 ^b^ (0–8)	10 ^c^ (0–14)	2 ^d^ (0–7)
**Fat exchanges**	1 (0–2)	1 ^a,b,c,d^ (0–1)	1 ^a^ (0–2)	0 ^b^ (0–1)	2 ^c^ (0–2.5)	0 ^d^ (0–1)

Values displayed are median and interquartile range per commercial portion of the product. Supplements with high volumes (>500 mL) were excluded from median (interquartile range) calculations. ^a^ Statistical difference between isocaloric and high-energy supplements. ^b^ Statistical difference between isocaloric and high-energy–high-protein supplements. ^c^ Statistical difference between isocaloric and high-energy–normal-protein supplements. ^d^ Statistical difference between isocaloric and high-protein supplements.

**Table 2 nutrients-14-03267-t002:** Spearman correlation coefficients between macronutrient content and calculated food exchanges.

		Carbohydrates (g)	Protein (g)	Fat (g)	Fiber (g)	Energy (kcal)
**Protein (g)**	Correlation Coefficient	0.287		0.454	0.211	0.586
	*p*	<0.001		<0.001	<0.001	<0.001
**Fat (g)**	Correlation Coefficient	0.524	0.454		0.238	0.835
	*p*	<0.001	<0.001		<0.001	<0.001
**Fiber (g)**	Correlation Coefficient	0.109	0.211	0.238		0.168
	*p*	0.02	<0.001	<0.001		0.001
**Energy (kcal)**	Correlation Coefficient	0.805	0.586	0.835	0.168	
	*p*	<0.001	<0.001	<0.001	0.001	
**Starch exchanges**	Correlation Coefficient	0.948	0.582	0.612	0.116	0.717
	*p*	<0.001	<0.001	<0.001	0.01	<0.001
**Lean meat exchanges**	Correlation Coefficient	0.512	0.787	0.575	0.208	0.415
	*p*	<0.001	<0.001	<0.001	<0.001	<0.001
**Fat exchanges**	Correlation Coefficient	0.718	0.565	0.829	0.201	0.737
	*p*	<0.001	<0.001	<0.001	<0.001	<0.001

Values of *p* < 0.05 were considered statistically significant.

**Table 3 nutrients-14-03267-t003:** Example of a 2100 kcal diet plan with incorporation of ONS.

	Initial Diet Plan (Foods Only) 2100 kcal for a 70 kg Older Adult					Final Diet Plan (Supplement + Foods) 2100 kcal for a 70 kg Older Adult	
						**Fortimel Energy Vanilla 2 × 200 mL ***	
	Food exchanges (without using ONS)	CHO	PROT	FAT	ENERGY	ONS Exchanges	Food Exchanges
Milk							
non-fat	0	0	0	0	0		0
medium fat	2	24	16	10	240		2
whole	0	0	0	0	0		0
Fruit	3	45	0	0	180		3
Vegetables	3	15	6	0	75		3
Carbs	10	150	30	3	800	5	5
Meat							0
very lean	0	0	0	0	0		0
lean	6	0	42	18	330	2	4
medium fat	0		0	0	0		0
high fat	0		0	0	0		0
Fat	10	0	0	50	450	4	6
Total		234	94	81	2075		

* Values for ONS exchanges are taken from Appendix A. Two hundred milliliters of product contains 36.8 g carbohydrates, 11.8 g protein and 11.6 g fat. ONS: oral nutritional supplement. Final food exchanges are calculated as follows: initial food exchanges—ONS exchanges. Both dietary plans provide 2100 kcal and approximately 234 g carbohydrate, 94 g proteins and 81 g fat. In this example, energy was calculated as 30 kcal/kg and protein was calculated as >1 g/protein/kg according to ESPEN recommendations for older adults [10]. This example is given for the understanding purposes of the reader. It should not be considered as a guide for patient consultation since diet plans are individualized.

## Data Availability

The data of the present study can be found in the attached Appendix A.

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
