# Peer review of "Development of a Tool for Determining the Equivalence of Nutritional Supplements to Diabetic Food Exchanges"

_nutrients, 2022, doi:10.3390/nu14163267_

Round 1
Reviewer 1 Report
The manuscript has been improved by the reviewer's comments and has thus benefited from a better reception. However, before publishing the manuscript in a journal, I propose to include minor corrections:
1. in the sub-section 'Categories of oral nutritional supplements', please indicate how many products were in the group; isocaloric supplements, high energy supplements etc. Indicate the quantity e.g.: in brackets.
2. Separate the 'Conclusion' section as it is in the journal requirements.
I recommend the paper for minor revision.
Author Response
We would like to thank the reviewer for his/her comments. We have done the proposed corrections and we include our answers point by point to the reviewer’s comments.
- in the sub-section 'Categories of oral nutritional supplements', please indicate how many products were in the group; isocaloric supplements, high energy supplements etc. Indicate the quantity e.g.: in brackets.
We have now included the number of products in each category (see lines 96-100 of the revised manuscript).
“Isocaloric supplements (with normal energy values) (n=150), high energy supplements (n=304), high energy- high protein supplements (n=302), high energy- normal protein supplements (n=303), and high protein supplements (n=173). It is noted that some supplements may fall in more than one categories.”
- Separate the 'Conclusion' section as it is in the journal requirements.
We have updated the manuscript accordingly.
Reviewer 2 Report
The authors assess the nutrient content for a total of 461 nutritional supplements and products and calculated the diabetic food exchanges. They also provided an excel for the calculation of food exchange of newly developed products. Overall, the study provided useful information for dietitians. Here are some comments and concerns.
1. Please change all the red font to black.
2. The introduction starts with sarcopenia, which leads the readers to think more specifically about the care of sarcopenia. However, after reading the aim and the result of the study, clearly sarcopenia is not the key point. Besides, malnutrition is not the most important risk factor for sarcopenia. I suggest the authors revise the introduction.
3. Line 38, please provide the full name of ESPEN.
Author Response
We would like to thank the reviewer for his/her comments. We have done the proposed corrections and we include our answers point by point to the reviewer’s comments.
- Please change all the red font to black.
The font has changed.
- The introduction starts with sarcopenia, which leads the readers to think more specifically about the care of sarcopenia. However, after reading the aim and the result of the study, clearly sarcopenia is not the key point. Besides, malnutrition is not the most important risk factor for sarcopenia. I suggest the authors revise the introduction.
The introduction has been updating focusing more on malnutrition (lines 29-32 of the revised manuscript).
“Malnutrition is an evident problem in hospitalized patients [1], …. along with dietary advice [7].”
- Line 38, please provide the full name of ESPEN.
We have now included the full name of ESPEN (see line 35 of the revised manuscript).
This manuscript is a resubmission of an earlier submission. The following is a list of the peer review reports and author responses from that submission.
Round 1
Reviewer 1 Report
The work of Detopoulou et al. addresses the design of a tool to calculate the equivalence of oral nutritional supplements to diabetic food exchanges.
Although the objective and idea of the present study is interesting, in my view the quality of the manuscript does not meet the expectations of Nutrients Journal. The Introduction section does not provide a sufficient background that properly contextualize the issue addressed. The design and working procedure followed in this study are not adequately described. Authors have not indicated the inclusion and exclusion criteria to select the companies and brands of oral nutritional supplements. Likewise, criteria to define the categories of oral nutritional supplements are not clearly included. The Results and Discussion sections do not provide a proper and comprehensive interpretation of the most important findings obtained in this work. Conclusions are unspecific and quite general. References are not properly present in the text and references section.
Based on all these premises, I suggest the rejection of the present manuscript.
Reviewer 2 Report
The reviewer appreciate the interest of the authors in the development of this manuscript. It is an interesting topic.
The main aim of the work was to develop a suitable tool for dietitians and health professionals to obtain: information on dietary supplements with corresponding estimated nutrient content and diabetic nutrient exchangers for each product. In addition, to obtain a suitable tool to use for any other ONS not included in the database, so that any current or future need could be met.
The manuscript is generally well written. However, the following issues have to be addressed before this manuscript is suitable for publication.
Authors should, in accordance with the requirements of the journal, remove the headings of each section of the description from the abstract, i.e.: Background, Methods etc.
Line: 69-78. Please indicate the reference to the excel file.
Line: 103. The correct notation for the significance level is p=0.05
Table 1. Please change the footnotes in the table to letter characters. No footnote explanation for the one sign.
Fig. 1. The title of the vertical axis overlaps the drawing.
In the title of Table 2, please add in brackets at what level of significance the correlations were tested.
The authors tested correlations at p=0.05 and a significance level of p<0.001 was included in the table...?
"Supplementary material Table 2_New" does not allow to test the tool created, no access.
The reviewer appreciate the interest of the authors in the development of this manuscript. Constructing a suitably helpful tool for dietitians and health professionals is by all means a good idea, however, there are some doubts about its use, due to the limitations mentioned by the authors. I suggest MAJOR REVISIONS.